# The geometry of masking in neural populations

Dario L. Ringach[1]*

The normalization model provides an elegant account of contextual modulation in individual neurons of primary visual cortex. Understanding the implications of normalization at the population level is hindered by the heterogeneity of cortical neurons, which differ in the composition of their normalization pools and semi-saturation constants. Here we introduce a geometric approach to investigate contextual modulation in neural populations and study how the representation of stimulus orientation is transformed by the presence of a mask. We find that population responses can be embedded in a low-dimensional space and that an affine transform can account for the effects of masking. The geometric analysis further reveals a link between changes in discriminability and bias induced by the mask. We propose the geometric approach can yield new insights into the image processing computations taking place in early visual cortex at the population level while coping with the heterogeneity of single cell behavior.

[1] Departments of Neurobiology and Psychology, UCLA, Los Angeles, CA 90095, USA. *email: dario@ucla.edu

I ndividual neurons in primary visual cortex respond to stimulation within restricted areas of the visual field, which define their classical receptive fields[1–3]. These responses can be modulated by contextual stimuli presented within the classical receptive field or in the surrounding regions[4–6]. Cross-orientation and surround suppression are two well-known examples of contextual modulation[5,7–21].

The role that contextual modulation plays in cortical function remains an open question. Some consider such interactions to be directly involved in image processing, such as the detection and enhancement of smooth, spatially extended contours[22–37]. Others argue that the fundamental goal of contextual modulation is to generate a sparse, efficient representation of natural images[6,38–45]. Distinguishing between these theories is not straightforward, as the their goals are not mutually exclusive[6].

Here we focus on how contextual modulation transforms the activity of neural populations. Contextual modulation has been studied extensively in single neurons, leading to the development of the influential normalization model[6,46,47]. Among several phenomena, this model explains contrast invariance—the finding that the shape of the tuning curve of individual neurons measured at different levels of contrast are scaled versions of each other with responses saturating at high contrasts. It also offers an account of how tuning curves scale in the presence of a mask that, when presented by itself, does not produce a response.

In a network composed of neurons with homogenous tuning functions and normalization signals these properties would generalize from single cells to entire populations. For example, if all neurons in a population are contrast invariant, and if they share the same contrast response function, then the direction of the population response will be invariant to the contrast of a visual stimulus[48].

Unfortunately, we know cortical neurons exhibit a wide range of tuning properties[49], contrast response functions[50–52], and normalization pools[53]. It is not entirely surprising, therefore, that in a population of heterogenous neurons the properties of single cell responses derived from the classic formulation of normalization, such as contrast and subspace invariance, do not generalize to the population response[48].

Here we show that, despite the heterogeneity of responses in a cortical population, we can nevertheless capture the effects of contextual modulation by a simple geometric transformation—at least for the case of masking. The findings suggest that a succinct mathematical description how neural populations behave under contextual modulation is possible, and that its characterization can shed light into the image processing computations performed by early visual cortex[54].

## Results

### Population responses in masked and unmasked conditions.
We measured the responses of neural populations in mouse primary visual cortex using two-photon imaging (Methods). Mice were head-restrained but otherwise free to walk on a rotating wheel. The visual stimulus consisted of two conditions (Fig. 1a). In the *unmasked condition*, a full-field sinusoidal grating was presented while its orientation changed linearly with time $\theta = \pi t / T$ with a period $T = 10$ s. This stimulus has previously been used to measure orientation maps[55]. In the *masked condition*, the same rotating stimulus was presented superimposed on top of a mask consisting of a sinusoidal grating oriented vertically. We estimated the spiking responses of neurons using a standard processing pipeline involving image registration, signal extraction, and non-linear deconvolution[56]. The periodic nature of the stimulus was evident in the temporal responses of cells (Fig. 1b), as neurons tuned to one orientation respond once per cycle. As

described in earlier studies[57], locomotion modulated the overall magnitude of responses in the population (Fig. 1b, shaded regions).

### Heterogenous responses of single cells in masked and unmasked conditions.
We computed the average response of neurons in the unmasked and masked conditions over the cycle of the stimulus (Fig. 1c, solid curves). The temporal responses were corrected by the mean stimulus-response delay (see Methods). After this correction, the temporal profile of the response can be interpreted as an estimate of the tuning curve of the neuron. The mask was always present at an orientation of 90° (Fig. 1c, dashed lines; subsequent figures omit the location of the mask to avoid clutter). The shaded areas represent the mean response ± 2 SEM computed over all the trials.

We observed a substantial heterogeneity of responses. Some cells were well tuned to orientation in the unmasked condition but were completely suppressed by the addition of the mask (Fig. 1c*a*). Others did not show such dramatic suppression, but responded with a scaled down version of their unmasked responses (Fig. 1c*b*)—a behavior consistent with the classic normalization model[6,46,58]. Some neurons showed little or no difference between the responses in the two conditions (Fig. 1c*c*). Another group saw their unmasked responses enhanced by the mask (Fig. 1c*d*). Finally, somewhat surprisingly, a set of neurons showed very weak or no responses in the unmasked condition but responded vigorously in the presence of the mask (Fig. 1c*e*)[48].

We studied the range of behaviors in single cells (Fig. 1c) by comparing the mean response of the *i*th neuron over the stimulation cycle between unmasked and masked conditions, which we denote by $\mu_u^i$ and $\mu_m^i$, respectively (Fig. 2a). There was a significant anti-correlation: the stronger a neuron responded in the unmasked condition the weaker its response was in the masked condition and vice versa ($n = 3920$, $r = -0.55$, $p = 5.6 \times 10^{-312}$). We refer to neurons at the extremes of the distribution of behavior as *grating* and *plaid* cells (Fig. 2a, shaded areas). These groups were formally defined as the cells attaining the 10% lowest (grating cells) and highest (plaid cells) ratios of $\log_2\left(\mu_m^i / \mu_u^i\right)$ (Fig. 2a, inset). These groups represent behaviors found at the extremes of a unimodal distribution (there was no evidence of discrete classes of neurons).

Grating and plaid cells had different preferred orientations. Grating cells were preferentially tuned to the orientation orthogonal to the mask in both conditions (Fig. 2b, left panels). In grating cells, the introduction of the mask scaled down the responses by about a third but did not affect tuning (note the different *y*-scales in Fig. 2b). This is the type of responses one might expect from the classic normalization model[46,58]. Plaid cells, on the other hand, where preferentially tuned to the orientation of the mask (90°) when probed with grating stimuli in the unmasked condition, although their responses were relatively weak (Fig. 2b, top right). Instead, and somewhat surprisingly, these cells responded robustly to the orthogonal orientation (0°) under the presence of the mask (Fig. 2b, bottom right)—in other words, they responded best when the stimulus was a plaid with orthogonal components.

### Geometry of contextual modulation in neural populations.
The data in unmasked and masked conditions can each be represented as a matrix where the columns represent the tuning function of each cell (Fig. 3a). To ease visualization, we ordered neurons by their preferred orientation. The rows of the matrix represent the population response to a given orientation. We denote the mean population responses (across cycles of the stimulus) as a function of orientation in the unmasked and masked conditions by $r_u(\theta)$

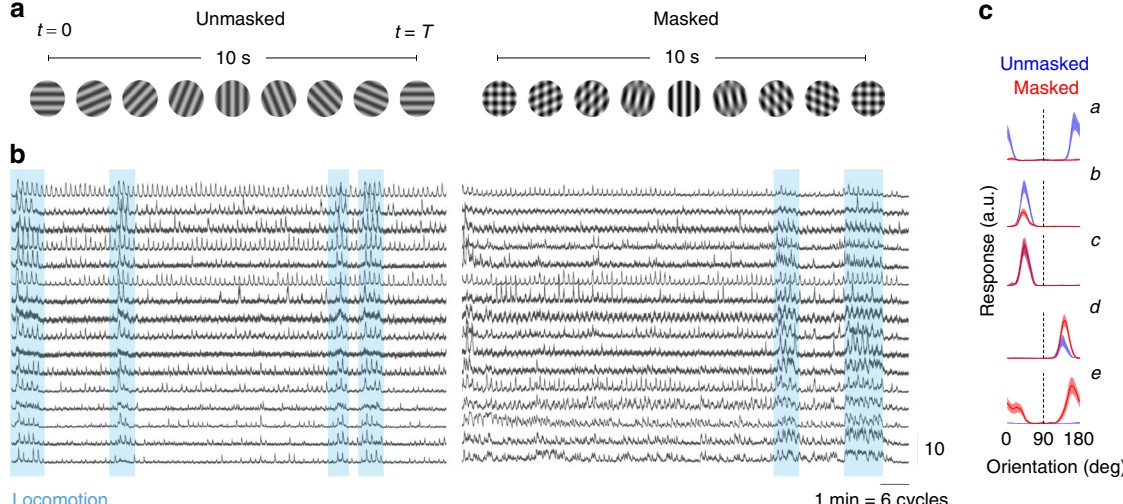

**Fig. 1** Measurement of population responses in masked and unmasked conditions. **a** Structure of the visual stimulus. Each of the lines show a single period of the stimulus in unmasked and masked conditions. **b** Examples of responses by individual neurons in both conditions. Periods of locomotion enhanced the overall responsivity of the population (shaded regions). Traces are plotted on a z-scored scale (vertical bar = 10). Horizontal bar represents 1 min of stimulation (or six periods of the orientation cycle). **c** Tuning in unmasked and masked conditions. Each trace shows the response of a neuron over the stimulation cycle after correction for neural delay, so they can be interpreted as a sweep of the orientation tuning curve of the neuron. The dashed line indicates the orientation of the mask. Blue traces represent the responses in the unmasked condition, while red traces represent responses in the masked condition. Shaded areas represent the mean response ± 2 SEM

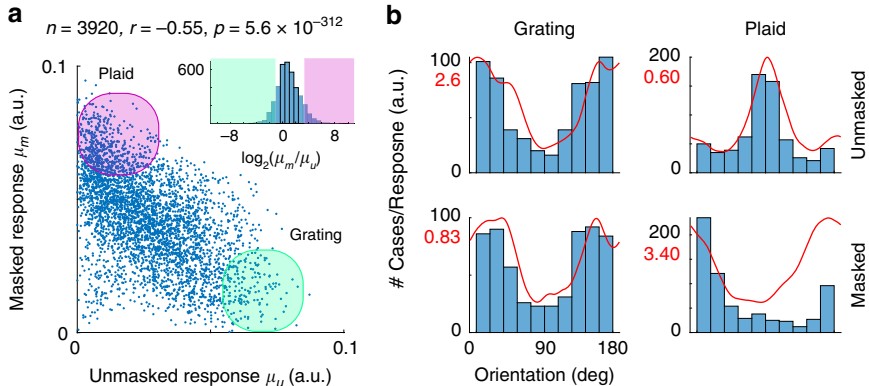

**Fig. 2** Characterization of responses in single neurons. **a** Anti-correlation between responses of neurons in masked and unmasked conditions. The mean responses of cells in the unmasked condition, $\mu_u$, are anti-correlated with the responses in the masked condition, $\mu_m$. The inset shows the distribution of $\log_2(\mu_m/\mu_u)$. Cells at the extreme of this distribution are termed grating (shaded green) and plaid (shaded pink) neurons. **b** Preferred orientation and average tuning of grating and plaid cells in unmasked and masked conditions. The histograms show the distribution of the preferred orientation of the neurons in each case. The red traces show the average tuning of neurons in each condition. The y-axis is labeled by cell count (in black) or by the amplitude of the responses (in red)

and $r_m(\theta)$, respectively. These vectors can be thought to describe parametric (closed) curves in a high dimensional space as $\theta \in [0, \pi]$ traverses the orientation domain (the dimension being the number of neurons in the population). We aim to understand the shape of these curves, the nature of the transformation $T : r_u(\theta) \to r_m(\theta)$ introduced by the mask, and how the outcome affects the discriminability of stimuli and biases the estimation of orientation in the masked condition.

We denote by $d_u(\theta, \varphi)$ the cosine distance between $r_u(\theta)$ and $r_u(\varphi)$ (Fig. 3b, left). The cosine distance is one minus the cosine of the angle between the two vectors. Because these vectors have positive entries representing a spike rate, the distance is bounded between zero and one. Similarly, we define $d_m(\theta, \varphi)$ as the cosine distance between $r_m(\theta)$ and $r_m(\varphi)$ (Fig. 3b, middle). As discussed below, the measurements $d_u(\theta, \varphi)$ and $d_m(\theta, \varphi)$ are related to the ability of the population to discriminate between two angles in

each condition. Finally, $d_{um}(\theta, \varphi)$ denotes the cosine distance between the population representation of $\theta$ in the unmasked condition and the representation of $\varphi$ in the masked condition. This measure captures the relative positions of the two curves which induces biases in the estimation of orientation in the presence of a mask. Namely, biases result when the structure of $d_{um}(\theta, \varphi)$ is not perfectly diagonal (Fig. 3b, right). We will denote the normalized population vectors by $\hat{r}_u(\theta) = r_u(\theta)/\|r_u(\theta)\|$ and $\hat{r}_m(\theta) = r_m(\theta)/\|r_m(\theta)\|$.

**Approximate orthogonality of signal and noise subspaces.** We selected the cosine distance as a metric because a substantial component of neural variability in the population response occurs along its direction[59]. To show this, we computed the mean and the covariance of the responses, $r_u(\theta)$ and $\sum_u(\theta)$. For each

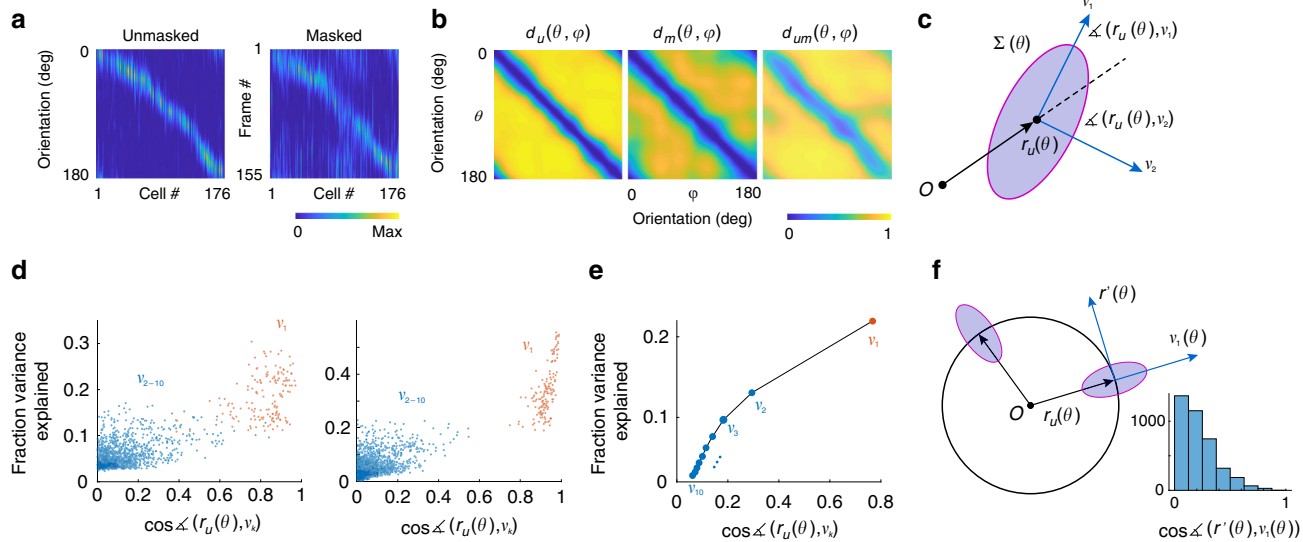

**Fig. 3** Characterization of population responses. **a** Responses of a population of neurons in the unmasked and masked conditions. Cells were ordered according to their preferred orientation, thus resulting in a diagonal structure. The rows for these matrices represent the population responses in the unmasked and masked conditions, $r_u(\theta)$ and $r_m(\theta)$. These curves describe a close curve as $\theta$ describes one cycle. **b** The intrinsic geometry of the curves is captured by the cosine distances between the representation of two orientations in the unmasked condition, $d_u(\theta,\varphi)$ (left panel), and masked condition, $d_m(\theta,\varphi)$ (middle panel). The relative positions of the curves with respect to each other is measured by the cosine distance between $r_u(\theta)$ and $r_m(\varphi)$, denoted by $d_{um}(\theta,\varphi)$ (right panel). **c** Simplified schematic showing the mean response to an orientation $r_u(\theta)$ along with its covariance matrix and the direction of the first and second eigenvectors, $v_1$ and $v_2$. **d** Results of two typical experiments showing the direction of the largest eigenvector (red dots) captured a substantial fraction of the of the variance (10–50%) and its direction is approximately aligned with the mean response (the cosine of the angle between the vectors was in the 0.8–1.0 range). Eigenvectors of higher rank (blue dots) accounted for significantly smaller fraction of the variance and their angles with respect to the mean response were much larger. **e** Population results. Average fraction of variance explained and the cosine of the angle with respect to the mean response averaged across all experiments. Error bars are about the size of the data points and not shown. **f** Simplified schematic in two-dimensions showing how the covariance of responses is anchored to the mean response of the vector population. The distribution of the cosine of the angle between $v_1$ and the direction of encoding $r'(\theta)$ has a mode at zero, meaning these vectors tend to be orthogonal to each other

orientation, we compared the direction of the population response with the direction of the eigenvectors, $v_k(\theta)$, of the covariance matrix (Fig. 3c) (we only present data for the ten eigenvectors with the largest eigenvalues). Data from individual experiments show that the first eigenvector, $v_1(\theta)$, (Fig. 3d, red dots) captures about 10–50% of the variance and its direction is close to that of the mean response, as evidenced by the cosine of the corresponding angle being in the 0.8–1.0 range. In contrast, eigenvectors of higher rank, $v_{2-10}(\theta)$ (Fig. 3d, blue dots) explain a significantly smaller fraction of the variance and their angles with respect to the mean response are much larger, as indicated by the cosine of these angles being typically around 0.3. The average relationship between explained variance and the cosine of the angle with respect to the mean response showed a clear dependence as a function of eigenvector rank (Fig. 3e).

The variability captured by the first eigenvector is due to fluctuations in behavioral state which modulates the magnitude of the response vector while leaving its direction relatively unchanged[57,60,61]. The direction of largest variability, $v_1(\theta)$, is approximately orthogonal to the direction of the encoding, $\hat{r}'_u(\theta)$ at all orientations (Fig. 3f). This feature of the covariance metrics justifies the adoption of the cosine metric and the hypothesis that the orientation of the stimulus is coded exclusively by the direction of the population vector[46,62].

**Multidimensional scaling of population responses**. To gain insight about the geometry of the curves and their relative positions we visualized them using multidimensional scaling adopting the cosine distance as a metric (Fig. 4). The curves represent the embeddings of $\hat{r}_u(\theta)$ (blue) and $\hat{r}_m(\theta)$ (red) in 3D space, while the spheres of matching colors indicate the point where the stimulus

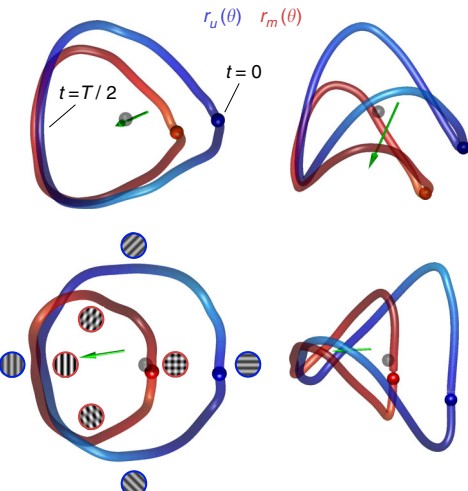

**Fig. 4** Multidimensional scaling (MDS) of population responses in unmasked and masked conditions. Each row shows two viewpoints of the result of one experiment. The curves were obtained by performing MDS simultaneously on the population responses in unmasked and masked conditions into 3D space using the cosine distance as a metric. The blue curve shows $r_u(\theta)$ and the red curve shows $r_m(\theta)$. The gray sphere represents the origin, and colored spheres represent the beginning of the cycle. The green arrows represent the shift in the white point between conditions. The stimuli represent the patterns at different locations on the curves for the two conditions (blue outline—unmasked condition, red outline—masked condition). The icons on the bottom left represent the configuration of the stimulus along different parts of the curves

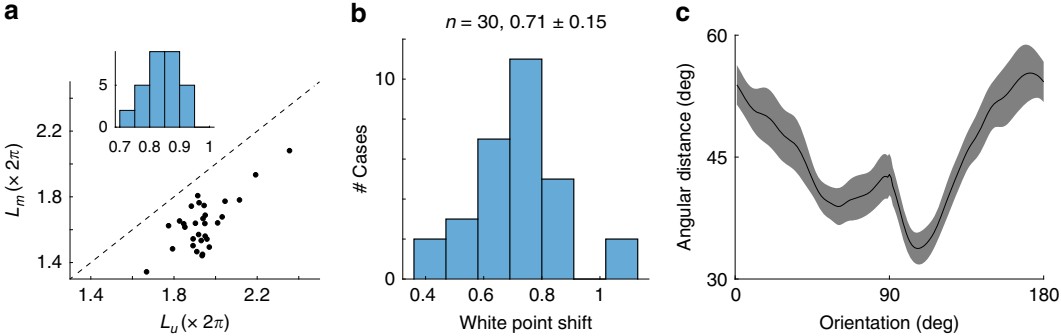

**Fig. 5** Basic geometric properties of population representations in unmasked and masked conditions. **a** Shrinkage of the length of the curves by the introduction of the mask. Scatterplot shows the lengths of the curves in unmasked ($L_u$) and masked ($L_m$) conditions. Dashed line represents the unity line. Inset shows the distribution of $L_m/L_u$ across all experiments. **b** Distribution of white-point shift ($\Delta$) across all experiments. **c** Measurements of the angle between $r_m(\theta)$ and the plane span$\{r_u(\theta), r_u(\pi/2)\}$ across all experiments. Solid line represents the mean, while the shaded area represents $\pm$ 2 SEM

cycle begins. We define the mean population response over the entire stimulation cycle as the *white point*, which we denote by denote by $\mu_u$ and $\mu_m$. The green arrows depict the shift of the white points between unmasked and masked conditions, with the stem of the arrow positioned at $\mu_u$ and the head at $\mu_m$. These examples are typical of what we observed in our experiments.

The shapes of $\hat{r}_u(\theta)$ and $\hat{r}_m(\theta)$ are similar, with the masked representation being a scaled down version of the original. The curves are farthest from each other at the beginning of the cycle, when the pattern in the masked condition consists of an orthogonal plaid and the one in the unmasked condition is a horizontal grating. The two curves are closest to each other near the middle of the cycle, when the pattern in the masked condition is a vertical grating with 100% contrast and the one in the unmasked condition is a vertical grating with 50% contrast. The curve $\hat{r}_m(\theta)$ appears to be rotated away from that of $\hat{r}_u(\theta)$, with the axis of rotation passing near the representation the mask. These features were consistent across our experiments suggesting that a scaling and rotation may explain the transformation of $\hat{r}_u(\theta)$ into $\hat{r}_m(\theta)$ induced by the mask. Of course, these visualizations ought to be interpreted with caution, as they are only approximate representations of the geometry of high dimensional objects. Thus, we must check these first impressions of the geometry by doing appropriate calculations in the native space.

**Masking shrinks and rotates population responses**. To verify our perception that curves are shrinking we computed their lengths[63], $L_u = \int_0^\pi \hat{r}'_u(\theta) d\theta$ and $L_m = \int_0^\pi \hat{r}'_m(\theta) d\theta$. The arguments represent the angular velocity at which the population changes its orientation and represent a measure of discriminability between nearby angles. The length, therefore, represents local discriminability summed over all orientations[63,64]. The mask had the effect of reducing the overall length of the curves by a factor of $0.84 \pm 0.05$ (mean $\pm$ 1 SD) (Fig. 5a). As we will soon demonstrate, this shrinkage is not uniform, but peaks near the orientation of the mask.

To verify our impression that the mask induces a change in the direction of the mean population activity, we defined the white-point shift as $\Delta = d(\mu_u, \mu_m)/((\rho_u + \rho_m)/2)$. Here, $\rho_u$ represents the average radius of the curve in the unmasked condition, calculated as $(1/\pi) \int_0^\pi d(r_u(\theta), \mu_u) d\theta$, and a corresponding definition applies to $\rho_m$. In other words, we measure the shift of the white point in terms of the average radius of the curves. Across the population we find $\Delta = 0.71 \pm 0.15$ (mean $\pm$ 1 SD)—a relatively large fraction (Fig. 5b), which is consistent with the

visualizations from multidimensional scaling. We will see this shift is important because it is partly responsible for generating biases in the estimation of orientation in the masked condition

**Rejection of the linear combination model**. With the geometric formalism in place, we can test a common model of population responses, which postulates that the response to a plaid can be written as a linear mixture of the population responses to the individual components[48,65]. The implication for our experiment is that $r_m(\theta) \in$ span$\{r_u(\theta), r_u(\pi/2)\}$ (recall the mask has orientation $\pi/2$). One way to test the prediction is to measure the angle formed by the vector $r_m(\theta)$ and the plane span$\{r_u(\theta), r_u(\pi/2)\}$. The results show a significant departure from the prediction, with angular deviations larger than 30° and significantly higher than zero ($p < 1.2 \times 10^{-4}$, bootstrap estimate, see Methods) (Fig. 5c). Thus, the data rule out the linear combination model, a finding that confirms and extends a prior result[48].

**Masking impairs discriminability and biases the decoding of orientation**. Next, we analyzed changes in discriminability induced by the mask. Discriminability between any two orientations depends both on the distance between the mean population vectors and the statistics of the noise. If the statistics of the noise are uniform in the sense that they translate with the direction of the population (Fig. 3f), we expect discriminability to be proportional to the distances $d_u(\theta, \varphi)$ and $d_m(\theta, \varphi)$. Nevertheless, given we have 90 cycles for each condition we were able to compute a proper *d-prime* measure for both masked and unmasked conditions, which we denote by $D_u(\theta, \varphi)$ and $D_m(\theta, \varphi)$ (see Methods) (Fig. 6a). To measure local discriminability (or just noticeable differences) we defined the threshold for detection in the unmasked condition $T_u(\theta)$ as the minimal angle $\Delta$ such that $D_u(\theta - \Delta/2, \theta + \Delta/2) \geq 4$ (Fig. 6a, iso-discriminability contours); we adopted a similar definition for the threshold in the masked condition, $T_m(\theta)$. Comparison of the thresholds in the two conditions revealed that the mask elevated thresholds around the orientation of the mask (at 90°) (Fig. 6a). Interestingly, the thresholds around the orientation orthogonal to the mask (0°) were not affected. A similar result is obtained if we perform a similar analysis based on $d_u(\theta, \varphi)$ and $d_m(\theta, \varphi)$ assuming uniformity of the noise (data not shown).

We then analyzed how the presence of the mask can lead to biases in estimates of orientation. We used a decoder based on population voting[62,67]. The estimated orientation was obtained as $\hat{\theta} = (1/2) \arg \int_{\theta=0}^\pi (1 - d_{um}(\theta, \varphi)) \exp(i2\theta) d\theta$. In other words, the population votes for each angle with a weight that depends on the

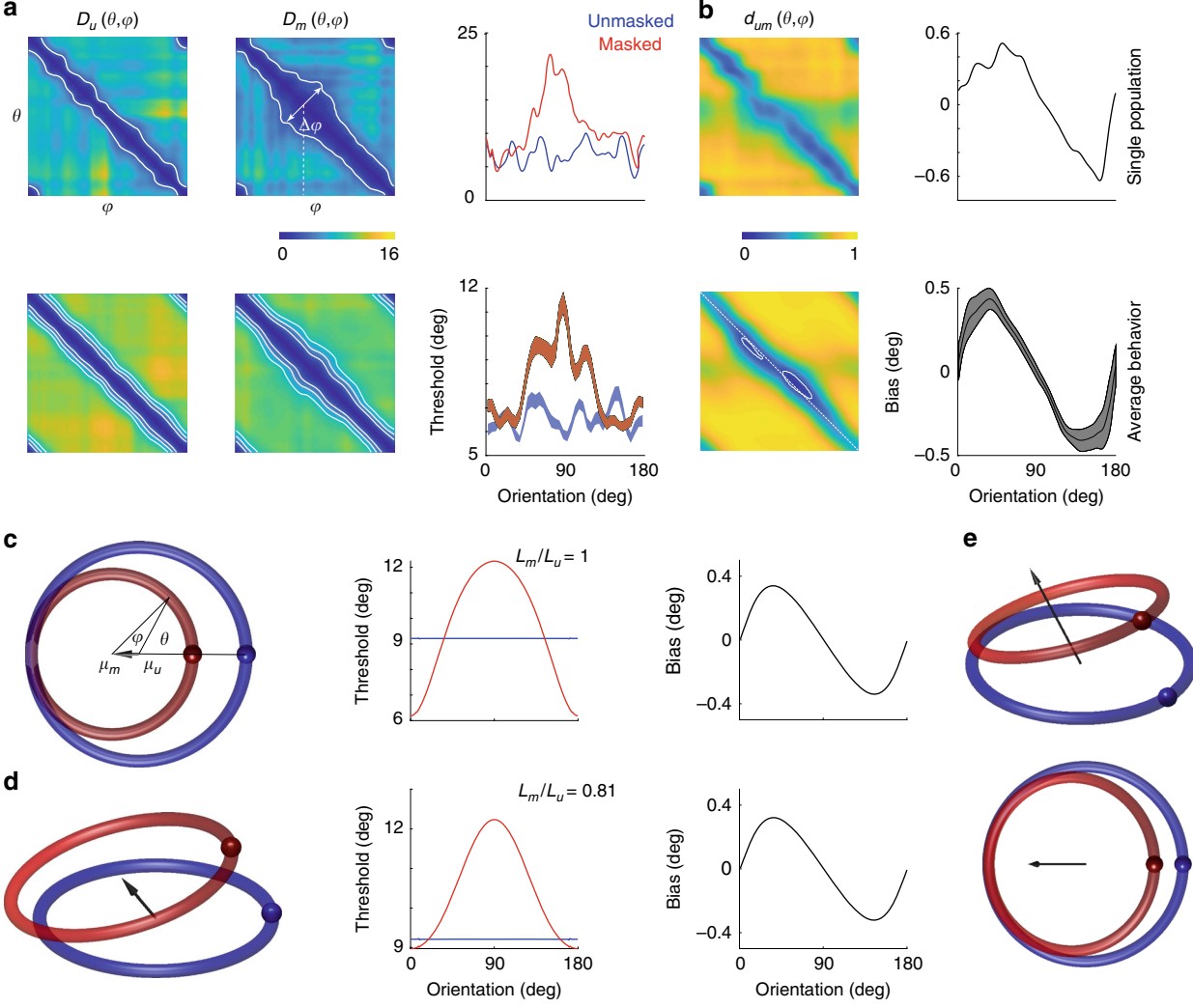

**Fig. 6** A geometric model of masking. **a** Discriminability (d-prime) between the representation of two orientations in unmasked (left panels) and masked (middle panels) conditions. The top panels show results for one experiment, while the ones at the bottom show the average across all our experiments. Iso-performance contour for the single experiment is shown at $d' = 4$. The iso-performance contours for the average behavior is shown at levels of $d' = 4, 6, 8$. The widening in the iso-performance contours in the masked condition reflect an increase in thresholds near the mask (which has an orientation of 90°). This is best shown in the panels on the right, which show the dependence of thresholds in masked (red) and unmasked (blue) conditions as a function of a base angle. In the average data the shaded areas represent ± 2 SEM. **b** Mutual distances and bias. Top panels show the mutual distance between orientations across masked and unmasked representations ($d_{um}$) and the expected bias from a decoder based on the distances. The non-diagonal structure of $d_{um}$ is more evident in the average data (bottom left panel), showing the locations of the minima of the main diagonal (white, dashed line). Bottom right panel shows the average bias across all our experiments. Shaded areas represent 2 SEM. **c** Two-dimensional geometric model of population coding[66]. The model assumes $r_u(\theta)$ and $r_m(\theta)$ are two circles in the plane. The displacement of their centers (white points) induce changes in the mutual distances inducing corresponding changes in threshold (middle panel) and bias (right panel). **d** The model can be extended by allowing displacement of the curves along a third dimension. **e** Two viewpoints of the same population activity in (**d**) but now normalized to yield $\hat{r}_u(\theta)$ and $\hat{r}_m(\theta)$

distance to the representation of each angle in the unmasked orientation—the smaller the distance the strongest the vote. The bias is then $b = (\hat{\theta} - \varphi) \bmod \pi$. We observe that except at the orthogonal orientation the estimates are biased towards the orientation of the mask (Fig. 6b). These biases arise because $d_{um}(\theta, \varphi)$ does not have a non-diagonal structure— the local minima of $d_{um}(\theta, \varphi)$ occur slightly off the main diagonal (Fig. 6b, bottom, white contours). Similar results are obtained using a simpler winner-takes-all decoder, where we pick $\hat{\theta} = \arg\min_\theta d_{um}(\theta, \varphi)$.

**A geometric model for population transformations under masking.** We tested if a simple geometric model[66], originally

developed to explain the effects of adaptation in psychophysical experiments, could also explain our masking data (Fig. 6c). The model assumes that in the unmasked condition the population response $r_u(\theta)$ describes a trajectory around the unit circle and that the effect of the mask is to translate and scale this response to yield $r_m(\theta)$. Translation is towards the population direction evoked by the mask, and the scaling is a typically a factor smaller than one. The model assumes that orientations are identified by the direction of the population vector, and that the decoder is unaware of the shift in the white point of the population between the two conditions. In other words, estimates of orientation are based on the direction of the population vector measured relative to the origin (which equals $\mu_u$ in this case) (Fig. 6c). The model has only two parameters, the magnitude of the shift of the white

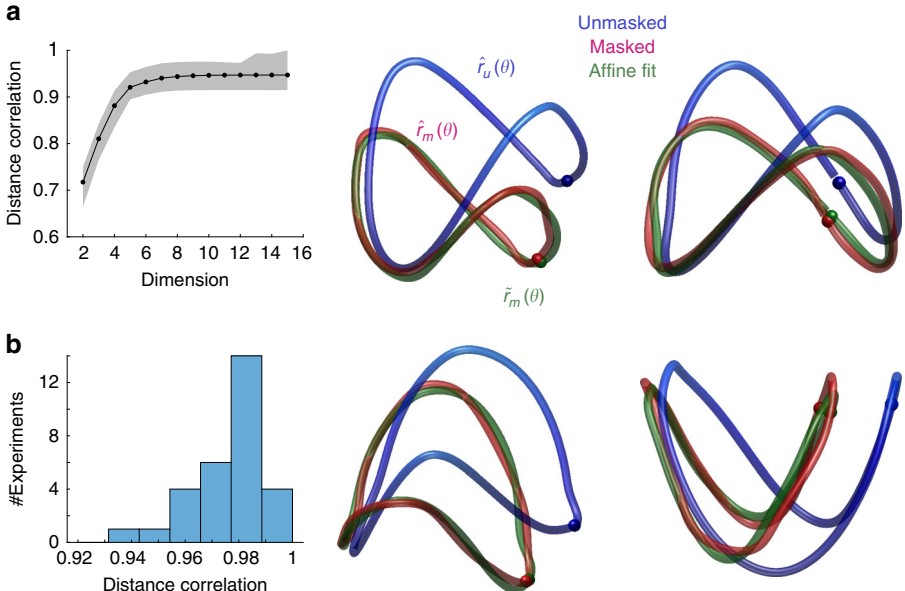

**Fig. 7** A simple geometric transformation accounts for the effects of masking in neural populations. **a** Multidimensional scaling indicates the data can be faithfully embedded in five dimensions. The y-axis represents the correlation between mutual distances in the native space and the low-dimensional embedding. Solid curve represents mean across all experiments; shaded area represent ± 2 SEM. **b** Fits of an affine model to low-dimensional representations of $\hat{r}_u(\theta)$ and $\hat{r}_m(\theta)$ in four different experiments. In each case, $\hat{r}_u(\theta)$ represents the population response in the unmasked condition (blue), $\hat{r}_m(\theta)$ represents the population response in the masked condition (red), and $\tilde{r}_m(\theta)$ is the best fit to the response in the masked condition by means of an affine transform. The curves are 2D projections of the five-dimensional fits. (**c**) Distribution of the correlation between the mutual distances between points in $\hat{r}_m(\theta)$ and corresponding points in the fit $\tilde{r}_m(\theta)$. The high correlation values agree with the visual impression in (**b**) that the fits are of excellent quality

point and a scaling factor. Its simplicity allows one to compute an analytical expression for both the threshold and the bias[66] (see Methods). Indeed, this model captures some of the behavior of observed in the data. First, it reproduces the dependence of threshold with orientation in the masked condition, showing a maximum centered around the orientation of the mask. Second, it reproduces the shape of the bias reflecting an attraction towards the orientation of the mask.

The model, however, fails in three fundamental ways. First, in the model, the population responses in both conditions lie within the same plane. As two independent vectors span the entire plane, it has to be the case that $r_m(\theta) \in \text{span}\{r_u(\theta), r_u(\pi/2)\}$ (so long as $r_u(\theta) \neq r_u(\pi/2)$). In other words, the responses ought to be explained by the linear mixture model[65]. However, we have already shown this is not the case in the data (Fig. 5d). This is also expected as the population activity in a homogeneous population with tuning curves other than cosine-shaped, cannot be embedded in 2D[63]. Second, in the model, both curves make a single revolution around the origin. This means that the lengths of the normalized responses are the same and equal to $2\pi$, predicting a ratio $L_m/L_u = 1$. Another way of stating this result is that both $\hat{r}_u(\theta)$ and $\hat{r}_m(\theta)$ are different parametrizations of the unit circle. However, the data show the ratios of the normalized lengths to be significantly less than one (Fig. 5a, tailed sign-test, $p = 9.3 \times 10^{-10}$). Third, the threshold is directly linked to how fast the population response changes its direction with orientation, which is given by $\|\hat{r}'_u(\theta)\|$ and $\|\hat{r}'_m(\theta)\|$. The faster the population direction rotates the lower the thresholds for discrimination. However, as we just pointed out the average across all orientations is constant under this model, $\int_0^{2\pi} \|\hat{r}'_u(\theta)\|d\theta = \int_0^{2\pi} \|\hat{r}'_m(\theta)\|d\theta = 2\pi$. This means that if the mask increases discriminability for some orientations it must decrease it for others[63]. This is reflected in the fact that the threshold in the masked condition fluctuates around the mean for the unmasked condition (Fig. 6c). The data, in contrast, indicates

that the effect of the mask is to impair the discriminability around the orientation of the mask, while there is little or no effect at the orthogonal orientation (Fig. 6a, right column). The data reject the prediction that increases in threshold at some orientations must be accompanied by decreases in threshold at other orientations (Fig. 6c). Fourth, we know from the analysis of single cell responses that some neurons are unresponsive in the unmasked condition but respond robustly in the presence of a mask (Fig. 2a). This fact alone indicates the population responses in the masked and unmasked conditions do not lie within the same subspace. We conclude that no model in 2D can explain the data.

Can the 2D model be extended to account for our results? One possible way to fix the model is to allow population activity to be embedded in dimensions higher than two. This will enable responses in the masked condition to move out of the subspace defined by the unmasked responses (Fig. 6d). To see if such an approach could work in principle, consider a toy example where responses in the unmasked condition lie on the unit circle whereas the one in the masked condition to be a the result of a transformation, $T(r) = \alpha A r + t$, where $A$ is an orthogonal matrix (representing a rotation), $\alpha$ is a scaling factor, and $t$ a translation. It is then possible to find parameters of the transformation that reproduce the ratio between the lengths of the curves, as well as the dependence of discriminability and bias on orientation (Fig. 6d, middle and right panels). A more general affine transformation can be represented in homogenous coordinates as $T(r) = Ar$ where the population vector now has an extra dimension to allow for translation. We can then write the transformation of the *normalized* population responses as $T(\hat{r}) = A\hat{r}/\|A\hat{r}\|$, which corresponds to a projective linear transformation[68] (Fig. 6e).

Based on these observations we fit the affine model to the data in individual experiments (see Methods). As a first step, we embedded the data in a lower dimensional space using

multidimensional scaling (Fig. 7a). On average, the correlation between mutual distances in the native space and the embedding saturated when the embedding had five dimensions, a selection we applied in the analyses of all the experiments. In each case, given population responses in the unmasked condition, we could find an affine transformation nicely accounted for the measured responses in masked condition (Fig. 7b). The quality of the fits can be evaluated by the correlation between mutual distances in the measured and predicted curves, which were higher than 0.92 in all our experiments (Fig. 7c).

## Discussion

Understanding how populations of neurons encode a physical attribute of a sensory stimulus, and how responses are transformed by contextual modulation is an important question in system neuroscience[54]. Here we considered the simpler problem of how the orientation of a sinusoidal grating is transformed by an additive mask.

At the single cell level, we observed a wide range of responses (Figs. 1c, 2). Interestingly, we found a group of neurons that do not respond to gratings in the unmasked condition but respond strongly to plaids in the masked condition (Fig. 2a). The maximal response of these *plaid neurons* occurs when the pattern is an orthogonal grating (Fig. 2b). Because this set of cells is only active in the masked condition their responses cannot be written as a linear mixture of their responses to gratings, as the neurons were not responsive to them. In other words, the responses in masked and unmasked conditions do not lie within the same subspace. This explains why the linear model (Fig. 5c) and the 2D geometric model (Fig. 6c) fail to account for the data. Grating and plaid cells are reminiscent of pattern and component cells[69,70]. Here, we use different terms because the definitions are not equivalent. We note, however, that the pattern index used to classify cells as pattern/component correlates with the plaid/ grating response we use here[71] and that mouse primary visual cortex contains a larger proportion of pattern cells than found in non-human primates[72]. Thus, we suspect that the neurons engaged during masking, that do not respond strongly to gratings in the masked condition, could represent pattern cells.

We observed that plaid cells, when probed with a single component in the unmasked condition, responded optimally (albeit weakly) to the orientation of the mask (Fig. 2b). While somewhat puzzling, the behavior in the masked condition might be explained if the addition of a grating orthogonal to a cell preferred orientation (as defined with single gratings) increases its response by releasing it from inhibition from oblique orientations in a ring model of orientation tuning[73].

The heterogeneous behavior of individual neurons makes it difficult to understand how the population behaves as single unit in unmasked and masked conditions. One approach to answering this question is to fit the normalization model individually to each cell and make sense of the distribution of model parameters and co-variation and their implications for the population. Here we used a geometric approach[48,66], which draws on ideas from the field of representational geometry[74,75], to study contextual modulation of neural populations directly. The analysis revealed that, despite a substantial heterogeneity in the behavior of individual cells, the map relating population responses in masked and unmasked conditions can be approximated as an affine transformation. When considering normalized responses, the corresponding map is a projective linear transformation[68]. The finding is so-far limited to masking, but we conjecture it may hold for other types of contextual modulation, such as interactions between the classical receptive field and the surround and sensory

adaptation. Indeed, a 2D model which accounts for psychophysical data on adaptation[66] (Fig. 6c) is an instance of an affine transform.

The geometric approach proved helpful in understanding several important properties of how population responses are modified by the introduction of a mask. First, it offered a rigorous test (and rejection) of a linear combination model[65]. The result can be understood as the mask having the effect of moving the population activity out of its original subspace. Second, the analyses revealed that the transformation cannot be a reparameterization of the same curve, of which the 2D model is a special case[66] (Fig. 6c). The reason is that all reparameterizations of the circle leave the length of the normalized curves invariant (thus predicting $L_m/L_u = 1$). In contrast, the mask was observed to shrink the length of the normalized representation (Fig. 5a). The shrinking of the normalized responses is not simply predicted by the known scaling of tuning curves in the normalization model. Third, we showed that the shift in the white point of the population is relatively large compared to the radius of the curve (Fig. 5c). This explains how a decoder which is unaware of such shift is bound to generate biased estimates. Finally, it clarified how a simple transformation can introduce changes in discriminability and bias in decoding (Fig. 6).

Our finding of a white-point shift appears to be at odds with the idea that adaptation keeps the mean population response invariant (population homeostasis)[76]. In our terminology, population homeostasis would have predicted that $\mu_u = \mu_m$, meaning no white-point shift. We suspect one reason for this discrepancy is rooted in the different stimuli used across studies. In the referenced study, a sequence of gratings with randomly chosen orientations was presented to the population. In one condition, the orientations were uniformly distributed; in the second condition, one orientation (the adapter) appeared more frequently than the others. In both conditions, any one stimulus consists of a single grating. It is possible that such design failed to engage the plaid cells that clearly play an important role in shifting the white point. Similarly, a previous report[65] selected cells to be analyzed only if their orientation tuning in response to a grating showed good selectivity (circular variance less than 0.85). Perhaps, plaid cells that were either unresponsive or weakly responsive to gratings failed to pass this criterion. The result would be biased towards gratings cells and it is possible that a linear combination model could be satisfactory when applied to this subpopulation of neurons. The shift in the white-point, however, is consistent with prior geometric models of psychophysical performance under adaptation or in the tilt illusion[66], and expected from the modulatory effect of masks on the tuning curve of individual neurons[66,77,78].

The affine class of transformations allows for a large variety of relationships between discriminability and bias. Surprisingly, the our data appear to conform with a recent theory[79] predicting a link between the threshold for discrimination and bias: $b(\theta) \propto \left(T(\theta)^2\right)'$. The relationship is derived under certain assumptions on efficient coding and decoding. As predicted, we find the extrema of the bias function aligning with regions of fast change in threshold, and the zero crossings of the bias aligning with the extrema of the threshold (Fig. 6). Thus, our data suggest that efficient encoding/decoding may be implemented by a simple geometric transformation of the population responses in early sensory areas.

Altogether the findings suggest that by analyzing the patterns of activity across large population of neurons we might be able to discover general principles of sensory representation, including topological[80] and geometrical transformations, that are undetectable at the single cell level. These patterns can allow us to

describe the transformations of representations in a simple way, as we demonstrated for masking, and shed light into more complex computations performed by cortical populations.

## Methods

**Animals**. All procedures were approved by UCLA's Office of Animal Research Oversight (the Institutional Animal Care and Use Committee) and in accord with guidelines set by the US National Institutes of Health. A total of 5 tetO-GCaMP6s mice (Jackson Labs), both male (3) and female (2), aged P35–56, were used in this study. Mice were housed in groups of 2–3, in reversed light cycle. Animals were naïve subjects with no prior history of participation in research studies. We imaged 30 different fields, and obtained data for 3920 cells, for a median of 111 cells per field (range: 50 to 275).

**Surgery**. Carprofen and buprenorphine analgesia were administered pre-operatively. Mice were then anesthetized with isoflurane (4–5% induction; 1.5–2% surgery). Core body temperature was maintained at 37.5 C using a feedback heating system. Eyes were coated with a thin layer of ophthalmic ointment to prevent desiccation. Anesthetized mice were mounted in a stereotaxic apparatus. Blunt ear bars were placed in the external auditory meatus to immobilize the head. A portion of the scalp overlying the two hemispheres of the cortex (approximately 8 mm by 6 mm) was removed to expose the underlying skull. After the skull was exposed it was dried and covered by a thin layer of Vetbond. After the Vetbond dried (approximately 15 min), it provided a stable and solid surface to affix the aluminum bracket with dental acrylic. The bracket was then affixed to the skull and the margins sealed with Vetbond and dental acrylic to prevent infections.

**Imaging and signal extraction**. Imaging was performed using a resonant, two-photon microscope (Neurolabware, Los Angeles, CA) controlled by Scanbox acquisition software (Scanbox, Los Angeles, CA). The light source was a Coherent Chameleon Ultra II laser (Coherent Inc, Santa Clara, CA) running at 920 nm. The objective was an x16 water immersion lens (Nikon, 0.8NA, 3 mm working distance). The microscope frame rate was 15.6 fps (512 lines with a resonant mirror at 8 kHz). We monitored locomotion using a rotary, optical encoder (US Digital, Vancouver, WA) connected to the rotation axel. The quadrature encoder was read by an Arduino board. We performed motion stabilization of the images, followed by signal extraction and deconvolution to estimate the spiking of neurons. The details of these methods are described elsewhere[56,60,81]. We used the average delay (387 ms) measured in reverse correlation experiments to correct for the stimulus-response delay in the data[81]. All cells that could be segmented were included in the analysis. We did not impose any criteria on inclusion—such as requiring neurons to be tuned to orientation. Segmentation was based on the temporal correlation of nearby pixels over time. This, of course, requires cells to be active to be segmented.

**Visual stimulation**. We measured the responses of neural populations in mouse primary visual cortex using two-photon imaging in tetO-GCaMP6s mice (Jackson Labs #024742). The visual stimulus consisted of two conditions. In the first, unmasked condition, a sinusoidal grating (50% contrast and a spatial frequency in the range 0.04–0.06cpd) was presented with an orientation that changed linearly with time $\theta = \pi t/T$, and a period $T = 10$ s. These parameters lead to 156 samples per period of the orientation cycle, as two-photon imaging was acquired at 15.6 fps. The spatial phase of the grating was updated every $T_\phi = 783$ msec by $\phi \leftarrow \phi + \pi/2 + n$, where $n$ was a random variable distributed uniformly $n \sim U(-\pi/8, +\pi/8)$. In other words, the grating underwent a "noisy contrast reversal" as its orientation changed continuously with time. This ensured that different spatial phases were present during different cycles of the stimulus. The unmasked condition was displayed for 15 min for a total of 90 cycles around the orientation domain. Immediately after, we added a vertical mask. The vertical mask also underwent a noisy contrast reversal with a period of 717 ms. A TTL pulse was generated by an Arduino board at the beginning of each stimulus cycle. The pulse was sampled by the microscope and time-stamped with the frame and line number being scanned at that time.

The screen was calibrated using a Photo-Research (Chatsworth, CA) PR-650 spectro-radiometer, and the result used to generate the appropriate gamma corrections for the red, green and blue components via an nVidia Quadro K4000 graphics card. The contrast of the stimulus was 99%. The center of the monitor was positioned with the center of the receptive field population for the eye contralateral to the cortical hemisphere under consideration. The locations of the receptive fields were estimated by an automated process where localized, flickering checkerboards patches, appeared at randomized locations within the screen. This measurement was performed at the beginning of each imaging session to ensure the centering of receptive fields on the monitor.

**Data analysis**. We computed discriminability between two angles $\theta$ and $\varphi$ as follows. Consider the responses in the unmasked condition. Let $r_u^i(\theta)$ be the response of the population in the $i$th cycle to a given orientation and let $\mu_u(\theta)$ be the mean population response across all trials. We define $d_u^i(\theta, \varphi) = d(\mu_u(\theta), r_u^i(\varphi))$.

We then compute the indices $F_\theta^i = (d_u^i(\theta, \theta) - d_u^i(\theta, \varphi))/(d_u^i(\theta, \theta) + d_u^i(\theta, \varphi))$ and $F_\varphi^i = -(d_u^i(\varphi, \varphi) - d_u^i(\varphi, \theta))/(d_u^i(\varphi, \varphi) + d_u^i(\varphi, \theta))$. Finally, we compute $D_u(\theta, \varphi)$ as the difference in the means of these distributions normalized by the average standard deviation. The same calculation was applied for the masked condition.

**Fitting the geometric model to experimental data**. Note that the affine model in $d$ dimensions has a total of $d(d + 1)$ parameters. Our data consists of $s = 155$ equally spaced samples (10 s period at 15.5 fps) of the continuous curves $r_u(\theta)$ and $r_m(\theta)$. Each sample provides $d$ constraints on the transform. Thus, we must have $d(d + 1) \leq ds$ or $(d + 1) \leq s$ to ensure the problem is not under-constrained. We handled this constraint by embedding the data in $R^5$ which faithfully represented the mutual distances in the dataset (Fig. 7a). We then fit the affine transform in this space. When visualizing the data (Fig. 7b), we plot 2D projections 2D the high dimensional curves. Similar results are obtained if, instead of multidimensional scaling, we use the first five principal components of the singular-value decomposition in the dimension-reduction step of the analysis.

**Analytic computation of threshold and bias**. In the simple geometric model of Fig. 6c it is possible to compute the threshold and bias. Consider a two-dimensional population code for orientation in the unmasked condition $r_u(\theta) = (\cos \theta, \sin \theta)$, which is transformed by a scaling and translation along the x-axis under the presence of the mask $r_m(\theta) = (a + b \cos \theta, b \sin \theta)$. Then, the velocity of $r_m(\theta)$ is

$$\|r_m'(\theta)\| = \frac{b(b + a \cos \theta)}{(a^2 + b^2 + 2ab \cos \theta)}$$

The threshold will be inversely proportional to the velocity $T_m(\theta) \propto 1/\|r_m'(\theta)\|$. Given a population direction in the masked condition, which in the plane is simply given by an angle $\varphi$, a decoder without knowledge of the white point shift will estimate the orientation by measuring the angle $\theta$ formed between the population vector with respect to $\mu_u$ (Fig. 6c), which a little geometry shows it is given by $\theta = \arctan((a + b \cos \varphi)/\sin \varphi)$. Thus, the bias is given by

$$\text{bias}(\varphi) = [\arctan((a + b \cos \varphi)/\sin \varphi) - \varphi]\mod 2\pi.$$

**Bootstrap estimate of angular deviations**. The population responses $r_u(\theta)$ and $r_m(\theta)$ were obtained as the average over 89 cycles of the stimulus (we discarded the first cycle to ensure the measurements were done in steady state). We want to test the null hypothesis $r_m(\theta) \in \text{span}\{r_u(\theta), r_u(\pi/2)\}$. To do this, we selected a random vector in the span and measured its angular distance from $\text{span}\{\breve{r}_u(\theta), \breve{r}_u(\pi/2)\}$, where $\breve{r}_u(\theta)$ represents the mean responses in a dataset obtained by randomly sampling trials (cycles) with replacement. This was done 50 times for each orientation. The distributions of angular distances were pooled over all orientations. We find that, under the null hypothesis, the probability of angular distances being larger than 30° is less than $p = 1.2 \times 10^{-4}$. All the values of the curve in Fig. 5c are above this level and, therefore, are significantly different than zero, thereby rejecting the null hypothesis.

**Reporting summary**. Further information on research design is available in the Nature Research Reporting Summary linked to this article.

## Data availability

Raw data containing spiking activity for each population in unmasked and masked conditions is available along with sample Matlab code the a Figshare repository: https://figshare.com/articles/The_Geometry_of_Masking/9922478

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

## Acknowledgements

I would like to thank Elaine Tring for help in surgical preparation and data collection. This work was supported by grant NIH R01 EB022915 to DLR.

## Competing interests

The authors declare no competing interests.

## Additional information

**Peer review information** *Nature Communications* thanks Aniruddha Das and other, anonymous, reviewers for their contributions to the peer review for this work. Peer review reports are available.

