## [Peer Review File · Nature Communications]

Reviewers' comments:

Reviewer #1 (Remarks to the Author):

In this manuscript, the author examines responses of mouse V1 neurons to oriented grating stimuli with and without a fixed superimposed grating. He finds the grating responses are modified by the superimposed grating, and uses multidimensional analysis and visualization to study the changes induced. The geometrical predictions and intuitions developed are interesting and a useful contribution, though in many cases the results described can be captured by previous analytical and mathematical work. I have a few general comments and concerns.

Major/overall:

- The paper is well written and clear. At times it is a bit overly brief and would benefit from additional explanation (primarily in Results) of what was done; some details below.
- Fig. 6 describes the main result: that an affine transformation can accurately capture the changes induced by a mask. But more quantification is needed, primarily quantification of performance of various models. L280 begins the discussion of this affine model. This is a nice way to think about the results and it is demonstrated graphically. But more should be done to capture how this model is better than e.g. the 2d model. Is its decoding performance better? If so, how much? How does this depend on various noise models? How does it depend on choices of the dimensions? Ideally this would be shown in figures.
- Section "mask impairs discriminability": "if the statistics of the noise are uniform..." Please discuss details of the statistics of the noise and what you can conclude from your data. More quantification of the noise statistics are needed, ideally in figures.
- Fig. 2A: Should explain in text why the greatest density of cells lie on a line of slope -1, while few cells lie near (0,0). How cells were selected for plotting should be explained in Results text.
- The white point shift can also be almost completely deduced from single cell responses, correct? Please expand in the manuscript.

Moderate/minor points

- "the temporal profile of the response can be interpreted as an estimate of the tuning curve of the neuron." Is 1C x-axis an explicit continuous temporal representation of the response? Or is it just spiking responses binned over specific periods of the stimulus? 1C should show data points and error bars, not just continuous lines. Also the way panel 1C is constructed should be explained in the Results text.
- L93: "plaid"
- L126: "in the sequel": awkward
- L141: should compute correlation between r_u and other eigenvectors, not just the first, for comparison
- L146: please explain "direction of the encoding" in more detail
- L155: green arrow?
- L269 "faster rotation speeds" - delete?
- some minor typo errors in citations, e.g. 49, 61

Reviewer #2 (Remarks to the Author):

Summary.

This manuscript offers a rigorous analysis of changes induced to the topology of population maps of orientation by the presence of a simultaneous masker. The development of the mathematical analysis is clear and well explained. However, it would be important to motivate the topological analysis more clearly and provide an argument for its relevance to the presumed visual tasks being performed.

Main

The author describes the responses (of mouse V1 neurons) to oriented gratings, either alone or in the presence of a vertical masker, in terms of the population vector in an N-dimensional space (N=the number of simultaneously recorded neurons). In both cases the trajectory of the population vectors is considered as a parametric curve in this N-dimensional space, parametrized by the orientation 'theta' of the test grating. From the topology of the curve, the author derives visual processing measures – such as the discriminability of test grating orientations (equated to the cosine distance between the relevant population vectors). The author also formally derives a transformation function that transforms the parametric curve with gratings alone, to that in the presence of the vertical mask.

These parametric curves are elegant and visually compelling. However the author needs to provide a clear argument motivating the choice of this approach. In fact, many of the points brought up by the author both here and in earlier work (Tring and Ringach, arXiv 2018) argue against the simple approach.

- The use of the population vector, and of transformation of this population vector with changed visual context presupposes a homogeneous population. But both Tring & Ringach (arXiv, 2018) and this manuscript argue against that, showing a marked heterogeneity in tuning (SF, orientation: in Tring & Ringach 2018) and response to a mask). So why is it important to formulate a single, homogeneous transformation matrix? Is it not more interesting to consider the heterogeneous nature of the transformation?

-

- Is it obvious that an important function of the visual system, even when faced with a 'plaid' would be to determine the orientation of the plaid components? It could be argued that while the grating is described by orientation, the plaid is described by 'texture' features: the angle between the elements that constitute the plaid and the size and pattern of the repeating element. Why is it important to evaluate how much the presence of the vertical masker changes the discriminability of the orientation of the 'test' grating component?

-

-

One way of addressing the issue is to alert the reader about the heterogeneity of the (mouse) V1 responses starting with the abstract and introduction. To say, not simply that "studying the geometry of neural populations can yield insights into the role of contextual modulation in the processing of sensory signals." but to summarize the heterogeneous effect that the contextual modulation has over the neural population making it necessary to move away from a single transformation matrix that can connect population response in the absence and presence of the masker. A succinct sense of this heterogeneity starting with the abstract may be the most interesting takeaway for the reader.

Minor points:

1: (Pg 4, line 108 etc): Specify that 'mean' population vector ($r_{\text{subscript}_u}(\theta)$) is averaged across cycles of stimulus presentation.

2: (Fig 3D, and Pg 6 line 131, 145 etc): The language at the end of the section 'Orthogonality of signal and noise subspaces' is too elegant and compact and can benefit by being expanded. It is clear that the primary direction of the noise / the principal eigenvector of the covariance matrix reflect changes in response gain and hence parallel to the population vector. But I needed to read the phrase "the direction of largest variability, $r_{\text{hat}}u(\theta)$, is orthogonal to the direction of the encoding, $r_{\text{hat}}u'(\theta)$, which is tangent to the unit sphere" multiple times before realizing that the 'encoding' is of theta and hence on the tangent, and thereby orthogonal to the noise.

3: Fig 3B: Since the sequence of orientation tuning peaks in the absence of masking (Fig 3A, left) is partially disrupted with masking (Fig 3A, right) the relationship of 'nearest neighbors' in the response space presumably changes with masking. I would have expected more clearcut examples of sharp island of low d' away from the diagonal, and of low d' at the diagonal. Is the absence of such islands just a result of smoothing?

4: Some typos. E.g. Fig 3 caption line 6. $D_{\text{subscript}_m}$ is referred to as the 'unmasked' condition; should be 'masked'.

Aniruddha Das

Reviewer #3 (Remarks to the Author):

This manuscript describes a novel way to describe the effect of stimulus manipulations on population responses. It compares the orientation tuning of V1 neurons with and without a mask, and neatly summarizes the effect of the mask on the entire population response with a carefully thought out geometrical analysis.

The results are novel, and the analysis is careful and thorough. Thus all of the major conclusions are fully justified.

My only comments concern how this relates to the earlier literature on masking. There are several places where it would be helpful to clarify where the findings here essentially recapitulate earlier findings, and where they show us something fundamentally new:

1) Their analysis rejects a simple linear model. But the idea that a linear model describes the interaction of two grating stimuli in V1 is long dead isn't it?

2) The white point changes. This is clearly different from the "homeostatic" mechanism, as they point out, but also as they point out the "adapter" is rather different here. It seems to me that the literature on cross-orientation inhibition would probably suggest white point changes (albeit not nearly so clearly demonstrated). Could they discuss this a little more?

3) The length changes. This seems to me something that could be anticipated from contrast gain control and cross-orientation inhibition, so again it would be useful to clarify how this relates to the earlier literature.

I'm glad the reviewers found the work of interest.

I hope the changes made in response to their constructive feedback has improved the manuscript and this revised version offers satisfactory answers to the questions raised.

All modifications from the original appear in **red** in the revised version of the manuscript.

The main changes made are as follows:

- I have **rewritten large sections of the manuscript**, including the introduction and discussion, to clarify some of the issues raised in the reviews.
- Following a suggestion by Reviewer #1, I have **added three new Figures** (Fig 3D, E and F) providing additional details about the structure of the covariance of responses and to better explain the relationship between the direction of the largest eigenvector and the “direction of encoding” (Reviewers #1 and #2).
- Following a suggestion by Reviewer #1, I have added **a new Figure 7** showing population results on the quality of the affine model fits across all our experiments.

A point-by-point reply to the reviewers' comments follows.

Reviewers' comments:

Reviewer #1 (Remarks to the Author):

In this manuscript, the author examines responses of mouse V1 neurons to oriented grating stimuli with and without a fixed superimposed grating. He finds the grating responses are modified by the superimposed grating, and uses multidimensional analysis and visualization to study the changes induced. The geometrical predictions and intuitions developed are interesting and a useful contribution, though in many cases the results described can be captured by previous analytical and mathematical work. I have a few general comments and concerns.

Major/overall:

- The paper is well written and clear. At times it is a bit overly brief and would benefit from additional explanation (primarily in Results) of what was done; some details below.

Thank you. I agree the descriptions were terse at points. I have expanded the manuscript to describe in more detail the procedures, findings and the conclusions section.

- Fig. 6 describes the main result: that an affine transformation can accurately capture the

changes induced by a mask. But more quantification is needed, primarily quantification of performance of various models. L280 begins the discussion of this affine model. This is a nice way to think about the results and it is demonstrated graphically. But more should be done to capture how this model is better than e.g. the 2d model. Is its decoding performance better? If so, how much? How does this depend on various noise models? How does it depend on choices of the dimensions? Ideally this would be shown in figures.

Following the reviewer's recommendation, we have now added a new Fig 7 where the quality of the fits of the affine model across the population is analyzed in more detail.

Regarding the 2D model, note that many of the conclusions of the study depend on geometric properties of the curves – such as their lengths and the angles formed by the responses. In particular, the rejection of the linear mixture model and the 2D encoding model are based on geometric properties of the responses alone. These conclusions do not depend on the decoding of orientation. I hope the updated text makes this point more clearly. Changes in discriminability and biases are also largely dependent on the geometric transformation of the curves and are robust to the choice of decoder. As stated in the Results, we obtained the same basic result for the population voting decoder and the winner-take-all decoder. The reviewer correctly points out that it may be possible to develop more accurate decoders by incorporating a more detailed noise model. However, our goal here was not to develop complex decoding methods, but to show how the geometric transformation of responses will necessarily give rise to changes in discriminability and induce biases for any reasonable decoder.

The size of the populations involved are stated in Methods/Animals. I imaged 30 different fields, and obtained data for 3920 cells, for a median of 111 cells per field (range: 50 to 275). The only significant effect I observed was a trend showing that the fraction of variance explained by the first eigenvector decreased with the size of the population (graph on the right). However, there are confounding factors that make the interpretation of this graph difficult. Namely, smaller populations were obtained at higher magnification factors in the microscope. As single cells comprised a higher number of

pixels (or samples) the signal-to-noise in these measurements were typically higher than those where the magnification and the number of pixels per cell was lower. Thus, a lower SNR for larger populations might explain the trend. One way to explore this in more detail would be to sub-sample populations of cells from the same recording and see if the effect remains. This will ensure different population dimensions were obtained with the same SNR levels. While this is a worthwhile endeavor, I feel this will require extensive computations that are beyond the scope of this paper.

- Section "mask impairs discriminability": "if the statistics of the noise are uniform..." Please

discuss details of the statistics of the noise and what you can conclude from your data. More quantification of the noise statistics are needed, ideally in figures.

Following the reviewer's suggestion, we have added new Figs 3D and 3E to quantify in more detail how the fraction of variance explained and the cosine of the angle between the eigenvectors and the mean response co-vary for the 10 eigenvectors with the largest eigenvalues. Fig 3D shows the result in individual experiments while Fig 3E shows the average dependence across all our experiments

- Fig. 2A: Should explain in text why the greatest density of cells lie on a line of slope -1, while few cells lie near (0,0). How cells were selected for plotting should be explained in Results text.

We now clarify in the methods that all cells that could be segmented were included in the analysis. We did not impose any criteria on inclusion – such as requiring neurons to be tuned to orientation. Segmentation was based on the temporal correlation of nearby pixels over time. This, of course, requires cells to be active to be segmented. Cells that do not respond in either the masked or unmasked condition would therefore not be included (these would be cells near (0,0)).

- The white point shift can also be almost completely deduced from single cell responses, correct? Please expand in the manuscript.

I am unsure as to how this is the case. The effect of the mask on single cell responses is heterogenous (Fig 1C) and the white point is defined as the average across all orientations of the *normalized* population vector – meaning it is a population property.

Moderate/minor points

- "the temporal profile of the response can be interpreted as an estimate of the tuning curve of the neuron." Is 1C x-axis an explicit continuous temporal representation of the response? Or is it just spiking responses binned over specific periods of the stimulus?

The stimulus undergoes a full orientation cycle every 10 seconds. The microscope sampled responses at 15.6 frames per sec. This means that we obtain a total of 156 samples of the population response in the 180 deg period of the orientation. The responses represent the population response sampled at those locations. We have now clarified this in the methods section.

1C should show data points and error bars, not just continuous lines.

As requested by the reviewer, we have added error bars to the panels in Fig 1C.

Also the way panel 1C is constructed should be explained in the Results text.

We have added text to the results and Fig 1C legend to explain the data being shown.

- L93: "plaid"

Thank you. Fixed.

- L126: "in the sequel": awkward

Fixed.

- L141: should compute correlation between r_u and other eigenvectors, not just the first, for comparison

New Figure 3E shows fraction of variance and correlation for all first 10 eigenvectors.

- L146: please explain "direction of the encoding" in more detail

I have rewritten this entire paragraph and added Fig 3F to clarify this idea.

- L155: green arrow?

Yes, thank you. Fixed.

- L269 "faster rotation speeds" - delete?

Thank you again. Deleted.

- some minor typo errors in citations, e.g. 49, 61

Fixed.

Reviewer #2 (Remarks to the Author):

Summary.

This manuscript offers a rigorous analysis of changes induced to the topology of population maps of orientation by the presence of a simultaneous masker. The development of the mathematical analysis is clear and well explained. However, it would be important to motivate the topological analysis more clearly and provide an argument for its relevance to the presumed visual tasks being performed.

Main

The author describes the responses (of mouse V1 neurons) to oriented gratings, either alone or in the presence of a vertical masker, in terms of the population vector in an N-dimensional space (N=the number of simultaneously recorded neurons). In both cases the trajectory of the population vectors is considered as a parametric curve in this N-dimensional space, parametrized by the orientation 'theta' of the test grating. From the topology of the curve, the author derives visual processing measures – such as the discriminability of test grating orientations (equated to the cosine distance between the relevant population vectors). The author also formally derives a transformation function that transforms the parametric curve with gratings alone, to that in the presence of the vertical mask.

These parametric curves are elegant and visually compelling. However the author needs to provide a clear argument motivating the choice of this approach. In fact, many of the points brought up by the author both here and in earlier work (Tring and Ringach, arXiv 2018) argue against the simple approach.

The use of the population vector, and of transformation of this population vector with changed visual context presupposes a homogeneous population. But both Tring & Ringach (arXiv, 2018) and this manuscript argue against that, showing a marked heterogeneity in tuning (SF, orientation: in Tring & Ringach 2018) and response to a mask). So why is it important to formulate a single, homogeneous transformation matrix? Is it not more interesting to consider the heterogeneous nature of the transformation?

Thank you for bringing up these fundamental questions.

We agree that neurons show heterogeneous responses to the mask and each can be reasonably be fit using the normalization model. Indeed, in the original 1992 paper, Heeger explained how suppression and enhancement of the responses by a mask could be explained by different choices of model parameters.

The question we ask is what does this mean for the population as a whole?

One approach is to try to understand the distribution of normalization model parameters that fit the population, try to elucidate how parameters (such as tuning and normalization pool) co-vary across neurons, and somehow try to make sense of such dependencies. Much of the work in this area has relied on such studies.

Here we take an alternative approach. Instead of modeling each cell individually, we ask whether the representation of the population can be captured directly using simple mathematical models. This may seem counterintuitive given the heterogeneity of the population. We know the populations we study have properties that differ not only on their preferred orientation, but also in their preferred spatial frequencies, spatial summation properties and other characteristics. The surprising finding from our study is that, despite such variability, the representation of the population in masked and unmasked conditions can be understand using a simple geometric

transformation. The findings are repeatable from one population to the next. Thus, despite a large heterogeneity at the single cell level, large populations behave in a surprisingly simple way.

We have modified the introduction and discussion sections to make these points clear.

Is it obvious that an important function of the visual system, even when faced with a 'plaid' would be to determine the orientation of the plaid components? It could be argued that while the grating is described by orientation, the plaid is described by 'texture' features: the angle between the elements that constitute the plaid and the size and pattern of the repeating element. Why is it important to evaluate how much the presence of the vertical masker changes the discriminability of the orientation of the 'test' grating component?

Two or more orientations are often present in natural images at the same location. Specifically, when the boundaries of objects at different depths overlap, corners, T or X junctions arise depending on the occlusion relationships of the participating objects. Much psychophysical data shows such locations are known to be perceptually important because they provide information to the visual system about how objects in the visual field stratify in depth, aiding in object segmentation and establishing the reflectance properties (including transparency) of different surfaces. In such cases, estimating the orientation of the participating boundaries is clearly important, as they assist in segmentation and the linking of contours belonging to the same object which has parts occluded by others. Plaids, of course, are a simplified class of stimuli that allow us to study the representation of multiple orientations. As pointed out in the seminal Adelson & Movshon paper, many conditions exist where the components of plaids do not cohere and, instead, are perceived as transparent surfaces, demonstrating the ability of the visual system to represent more than one visual attribute at any one point in the visual field.

One way of addressing the issue is to alert the reader about the heterogeneity of the (mouse) V1 responses starting with the abstract and introduction. To say, not simply that "studying the geometry of neural populations can yield insights into the role of contextual modulation in the processing of sensory signals." but to summarize the heterogeneous effect that the contextual modulation has over the neural population making it necessary to move away from a single transformation matrix that can connect population response in the absence and presence of the masker. A succinct sense of this heterogeneity starting with the abstract may be the most interesting takeaway for the reader.

Thank you. The abstract has been rewritten to emphasize the heterogeneity at the single cell level and to clarify the contribution of the geometric approach.

Minor points:

1: (Pg 4, line 108 etc): Specify that 'mean' population vector ($r_{u(\theta)}$) is averaged across cycles of stimulus presentation.

We now specify this in the text.

2: (Fig 3D, and Pg 6 line 131, 145 etc): The language at the end of the section 'Orthogonality of signal and noise subspaces' is too elegant and compact and can benefit by being expanded. It is clear that the primary direction of the noise / the principal eigenvector of the covariance matrix reflect changes in response gain and hence parallel to the population vector. But I needed to read the phrase "the direction of largest variability, $\hat{r}u(\theta)$, is orthogonal to the direction of the encoding, $\hat{r}u'(\theta)$, which is tangent to the unit sphere" multiple times before realizing that the 'encoding' is of theta and hence on the tangent, and thereby orthogonal to the noise.

Thank you. Reviewer #1 also had difficulty with this sentence. I have now rewritten this entire paragraph and added a Figure to explain this concept.

3: Fig 3B: Since the sequence of orientation tuning peaks in the absence of masking (Fig 3A, left) is partially disrupted with masking (Fig 3A, right) the relationship of 'nearest neighbors' in the response space presumably changes with masking. I would have expected more clearcut examples of sharp island of low d' away from the diagonal, and of low d' at the diagonal. Is the absence of such islands just a result of smoothing?

The changes are not so obvious to the naked eye, but note the maximum distance achieved is smaller for the masked condition (see the darker yellowish tones off the diagonal in the masked vs unmasked condition) and observe that the "thickness" of the blue diagonal near the center is a little bit larger for the masked condition when compared to the unmasked condition. However, these changes are made explicit later when we plot iso-discriminability contours in individual examples and the average across the population (Fig 6A).

4: Some typos. E.g. Fig 3 caption line 6. $D_{\text{subscript}_m}$ is referred to as the 'unmasked' condition; should be 'masked'.

Fixed.

Reviewer #3 (Remarks to the Author):

This manuscript describes a novel way to describe the effect of stimulus manipulations on population responses. It compares the orientation tuning of V1 neurons with and without a mask, and neatly summarizes the effect of the mask on the entire population response with a carefully thought out geometrical analysis. The results are novel, and the analysis is careful and thorough. Thus all of the major conclusions are fully justified.

My only comments concern how this relates to the earlier literature on masking. There are several places where it would be helpful to clarify where the findings here essentially recapitulate earlier findings, and where they show us something fundamentally new:

1) Their analysis rejects a simple linear model. But the idea that a linear model describes the interaction of two grating stimuli in V1 is long dead isn't it?

Note that we are not referring to a purely linear model, but a linear mixture model. Specifically, the model postulates that given two stimuli A and B , with contrasts α and β , the population response to the mixture $\alpha A + \beta B$ has the form $w_A(\alpha, \beta)r_A + w_B(\alpha, \beta)r_B$. Here, r_A and r_B are the population responses to the individual stimuli, while $w_A(\alpha, \beta)$ and $w_B(\alpha, \beta)$ are weights that are non-linear functions of the contrasts. Thus, the full model is nonlinear and allows one to explain averaging and winner-take all behavior seen for different combinations contrasts. This model has been proposed to explain the behavior of populations in cat area V1 (see cited ref 64) and certainly not "dead".

2) The white point changes. This is clearly different from the "homeostatic" mechanism, as the point out, but also as they point out the "adapter" is rather different here. It seems to me that the literature on cross-orientation inhibition would probably suggest white point changes (albeit not nearly so clearly demonstrated). Could they discuss this a little more?

Indeed -- we now point out this agreement in the Discussion section.

3) The length changes. This seems to me something that could be anticipated from contrast gain control and cross-orientation inhibition, so again it would be useful to clarify how this relates to the earlier literature.

Our discussion centers on the length of *normalized* responses. Prior work has clearly shown the *amplitude* of responses tend to be suppressed by the presence of a mask. The two are not the same. Take for example a simple 2D model where orientation is initially coded by the population vector describing a circle, then masking is expected to transform the population response into a translated ellipse. After normalization, however, this ellipse becomes a circle again. In other words, the two curves in unmasked and masked conditions are different parameterizations of the circle. Thus, despite changes in *amplitudes* of the responses we might predict from normalization, the length of the *normalized* curves is invariant and equal to 2π in both cases. This is one of the properties we use to reject the 2D geometric model. We have rewritten parts of the results and discussion to make this point clear.

REVIEWERS' COMMENTS:

Reviewer #1 (Remarks to the Author):

The author has addressed my concerns.

The additional figure (Fig. 7) exploring the performance of the model with additional dimensions is one way of addressing the quality of various kinds of models. It would be nice if these results (the difference between 2D and affine fit quality, and why affine performs better, incl. add'l dimensions) were mentioned in the abstract, but this is just my preference and is not necessary.

minor points:

- + add legend for Fig. 7B that says "unmasked data ", "masked data", "affine model fit"?
- + When I asked about the white point shift being deducible from known single cell responses, I was asking the same question as R3 pt 2. The text added to discussion (line 423, revised manuscript) addresses my question.

Reviewer #2 (Remarks to the Author):

No additional comments

Reviewer #3 (Remarks to the Author):

I am happy with the revised version and have no further comments.

I am glad the reviewers were largely satisfied by the revisions made to the paper.

Only Reviewer #1 had additional comments.

Reviewer #1 (Remarks to the Author):

The author has addressed my concerns.

I am glad the revised version addressed the concerns.

The additional figure (Fig. 7) exploring the performance of the model with additional dimensions is one way of addressing the quality of various kinds of models. It would be nice if these results (the difference between 2D and affine fit quality, and why affine performs better, incl. add'l dimensions) were mentioned in the abstract, but this is just my preference and is not necessary.

We have modified the abstract to mention the affine model. Describing the differences between the 2D and affine model in the abstract is not possible due to space limitations.

minor points:

+ add legend for Fig. 7B that says "unmasked data ", "masked data", "affine model fit"?

We added the legend as requested.

+ When I asked about the white point shift being deducible from known single cell responses, I was asking the same question as R3 pt 2. The text added to discussion (line 423, revised manuscript) addresses my question.

Thank you. I am glad this point has been clarified as well.

Reviewer #2 (Remarks to the Author):

No additional comments

Reviewer #3 (Remarks to the Author):

I am happy with the revised version and have no further comments.